# Staying Dry and Clean: An Insect’s Guide to Hydrophobicity

**DOI:** 10.3390/insects14010042

**Published:** 2022-12-31

**Authors:** Elizabeth Bello, Yutao Chen, Marianne Alleyne

**Affiliations:** 1Department of Entomology, University of Illinois at Urbana-Champaign, Urbana, IL 61801, USA; 2Program in Ecology, Evolution and Conservation Biology, University of Illinois at Urbana-Champaign, Urbana, IL 61801, USA; 3Beckman Institute for Advanced Science and Technology, University of Illinois at Urbana-Champaign, Urbana, IL 61801, USA; 4Department of Mechanical Science and Engineering, University of Illinois at Urbana-Champaign, Urbana, IL 61801, USA

**Keywords:** cuticle, surface topography, wettability, anti-wetting, hydrophobicity, self-cleaning, bioinspiration, hierarchical materials

## Abstract

**Simple Summary:**

Insects possess microscopic cuticular surface structures of different magnitudes. Such nano-, micro-, macro- and hierarchical structures often result in multifunctionality. In this review, we focus on hydrophobicity of the insect cuticle, since it often gives rise to other functions such as self-cleaning, anti-fogging, and anti-microbial activity. To do this, we reviewed the scientific literature on hydrophobic and superhydrophobic structures in insects. We found many insects possess unique structures and surface chemicals that make the cuticle waterproof. Among the many examples, we selected a few prominent ones to show the contribution of different levels of cuticular structures, as well as chemistry, in achieving hydrophobicity. We also discuss some instances of modern insect-inspired hydrophobic engineering designs. We show that insects are a great reservoir of inspiration for the guided design of novel materials with hydrophobic functionalities. Moreover, we also impart valuable insights on how material surfaces are important for biological systems.

**Abstract:**

Insects demonstrate a wide diversity of microscopic cuticular and extra-cuticular features. These features often produce multifunctional surfaces which are greatly desired in engineering and material science fields. Among these functionalities, hydrophobicity is of particular interest and has gained recent attention as it often results in other properties such as self-cleaning, anti-biofouling, and anti-corrosion. We reviewed the historical and contemporary scientific literature to create an extensive review of known hydrophobic and superhydrophobic structures in insects. We found that numerous insects across at least fourteen taxonomic orders possess a wide variety of cuticular surface chemicals and physical structures that promote hydrophobicity. We discuss a few bioinspired design examples of how insects have already inspired new technologies. Moving forward, the use of a bioinspiration framework will help us gain insight into how and why these systems work in nature. Undoubtedly, our fundamental understanding of the physical and chemical principles that result in functional insect surfaces will continue to facilitate the design and production of novel materials.

## 1. Fundamentals of Wettability in Nature

### 1.1. Wettability 

The wettability of synthetic and natural structured surfaces has undergone active investigation in recent years. Wettability characterizes a surface’s ability to get wet or the ability of a liquid to spread across a surface. This is impacted by both surface morphology and chemistry [1]. However, surface-structure-induced hydrophobicity is the more powerful mechanism. Smooth surfaces generally fail to repel water unless accompanied by hydrophobic surface chemistry. The maximum contact angle of water on a chemically treated smooth surface is roughly 120° [2]. In contrast, rough surfaces can exhibit excellent de-wetting properties and enhance hydrophobicity well beyond a 120° contact angle [3,4,5,6]. 

There are two models which characterize the wettability of a rough surface: the Cassie–Baxter model and the Wenzel model [7,8]. The Cassie–Baxter model includes a nonhomogeneous, three-phase liquid–air–solid interface [7]. In this model, air pockets are trapped between water droplets and the structured surface (Figure 1) [9,10]. The Wenzel model is a homogeneous, two-phase regime featuring a liquid–solid interface where water droplets penetrate the surface structure and are pinned at the surface (Figure 1) [8]. The hydrophobic nature of a material in this case is enhanced mainly due to an increase in surface area from surface roughness. 

Calculating the contact angle between the water droplet’s perimeter and the material’s surface is one method used to characterize wettability. The contact angle can range from 0° (superhydrophilic) to 180° (superhydrophobic). Contact angles between 0° and 90° are hydrophilic, while contact angles between 90° and 150°, and 150° and above, are characterized as hydrophobic and superhydrophobic, respectively (Figure 2). The advancing contact angle describes the contact angle between the droplet and the surface as water is applied while the receding contact angle describes the contact angle between the droplet and the surface as water is removed or evaporates. A surface that demonstrates droplets in the Wenzel state may be characterized as hydrophobic, but superhydrophobic characterization only occurs on surfaces with droplets in the Cassie–Baxter state [11]. 

Another important parameter is contact angle hysteresis (CAH), which is the difference between the advancing and receding contact angles. Droplets in the Cassie–Baxter state will tend to have large advancing and receding contact angles and a low contact angle hysteresis. Droplets in the Wenzel state will tend to have a large advancing contact angle, but a low receding contact angle, and thus a high contact angle hysteresis [11]. It is important to take hysteresis into consideration because two materials may have similar advancing contact angle measurements, but different receding contact angles and contact angle hysteresis, which drastically affect the adhesion and self-cleaning properties that are usually associated with superhydrophobic materials [5].

### 1.2. Hydrophobic Mechanisms in Nature

A variety of organisms exhibit hydrophobicity. The lotus leaf, for example, is among the most studied plant surfaces in nature; it is superhydrophobic with self-cleaning abilities, which has been appropriately named the “lotus effect” [12,13,14]. The superhydrophobic nature of the lotus leaf is caused by the combination of hierarchical structures which trap air underneath the water droplets, as well as surface waxes that are inherently water-repellent [13]. Other superhydrophobic plant surfaces include rice leaves, taro leaves, India canna leaves, *Salvinia* leaves, rose petals, and pitcher plants [9,12,15,16,17,18].

Some vertebrates also have hydrophobic capabilities. Two well-studied examples include bird feathers and gecko feet [19,20,21]. Ducks and feral rock pigeons use multiscale feather branching structures, along with natural preening oils, to induce hydrophobicity [19,20]. Gecko feet are covered with microscopic setae that branch further into nanoscopic spatula, enabling them to be superhydrophobic but also highly adhesive due to van der Waals interactions and capillary forces [21]. Gecko feet are covered with microscopic setae that branch further into nanoscopic spatula, enabling them to be superhydrophobic but also highly adhesive due to van der Waals interactions and capillary forces [21].

### 1.3. Hydrophobic Mechanisms in Insects

Insects are one of the most abundant groups of animals on the planet, thriving under a multitude of environmental conditions. Insects have adapted to almost all types of environments, with species living in polar regions [22,23] and other extreme conditions, such as arid deserts [24,25] and high altitudes [26]. Insects’ successful adaptations give rise to a myriad of specialized structures that have enabled their survival. 

Insect cuticle is a remarkable material that has long captured the attention of scientists. The cuticle can be thin and flexible, thick and rigid, smooth or rough, and can provide various functionalities such as adhesion, chemical sensing and defense, color manipulation, locomotion, mechanosensation, sound production, thermoregulation, (anti)reflectivity, and (anti)wetting [1]. It can also vary between sexes, life stages and body parts, and can change based on the environment in which the insect lives [27,28]. The surface of the cuticle can be equally as varied. Surface topography generally refers to any deviations from a perfectly flat plane at the surface of a material. In insects, the cuticle can be smooth in some areas and heavily patterned and textured in others and varies between species, sexes, life stages, and body parts.

Sclerotization is a process seen in insect cuticle formation where endogenously generated catecholamine derivatives react with structural proteins and chitin fibers [29,30,31]. This process typically occurs after molting and metamorphosis and can result in altered surface topography. Sclerotization may also be accompanied by melanization, a process that is essential to the tanning and hardening of the cuticle and is mediated by quinones that cross-link cuticular proteins [30]. As the outermost layer, the epicuticle is formed, but its surface topography is influenced by underlying cuticular layers. Overall, these chemical processes associated with cuticle formation create surface sculpturing that can result in structures such as air-retaining plastrons, diffraction gratings, and frictional surfaces [32,33]. These surface sculptures can take the form of regular- or irregular-spaced repeating patterns of scales, ridges, or hair-like structures found on the wings of butterflies [34], beetles [35], and flies [36], or more complex shapes such as the polygonal patterning found on springtails [37,38]. Because of the astounding diversity of insects and their cuticular patterning, there is a seemingly limitless wealth of resources to inspire the design and creation of novel materials with functional properties. 

While insects inhabit various environments, one commonality they share is interaction with water. Aquatic and semi-aquatic insects have developed sophisticated features to help them repel water. The body and legs of water striders possess micro-hair-like structures which help them rest and walk smoothly on water [39]. Certain bugs, such as backswimmers, have also developed characteristics to help them retain air bubbles, known as physical gills or plastrons, to breathe while underwater. A retained air layer located near wing microstructures may help insects detect water pressure changes to avoid predation [40]. The wings of some fly species are also hydrophobic, having structures at different length scales that collectively help them repel water in moist environments [36]. Alternatively, insects that live in dry areas have evolved similar features, but instead of repelling water, their cuticular structures help collect water. For example, the combination of waxy and hydrophobic, and non-waxy and hydrophilic, regions on the elytra of Namib desert beetles, combined with certain behaviors, allows them to capture water from fog layers more efficiently [41].

The cuticular surface of insects generally possess nanostructures, microstructures, or a combination of structures at different length scales, resulting in hydrophobic properties [11]. The textured surfaces suspend water droplets on top of these structures in the Cassie–Baxter state where there is an air-pocket trapped underneath the droplet, preventing the surface from getting wet as previously mentioned. Their (super)hydrophobicity can give rise to many nuanced and useful properties such as cooling, anti-fogging, anti-icing, and self-cleaning, etc., which can be utilized in a wide array of wetting- and de-wetting-related applications [11]. Rough hydrophobic surfaces enhance dropwise condensation, thus enhancing heat transfer and cooling [42,43,44]. Properties such as anti-fogging and anti-icing can be seen when the surface remains hydrophobic even during fog, high humidity, or low temperatures [45,46,47]. Self-cleaning characteristics occur when a water droplet, instead of wetting the surface, retains its almost spherical shape, and when subjected to a small, tilted angle, the droplet will trap dirt particles or other contaminants and roll off the surface, leaving it clean [48]. 

## 2. Terminology and Methodology 

### 2.1. Overview of Length Scale and Cuticular Terminology 

The most common cuticular structures resulting in multifunctionality exist on the length scales of nano-, micro-, and macrometers. Here, we define nanostructures as having at least one dimension (e.g., length, width, height) less than 1 μm. We define microstructures as having dimensions between 1 μm and 1 mm, and macrostructures as having dimensions larger than 1 mm. The term hierarchical is then used to describe an array of features in a given area with multiple length scales. It can also be used to describe a singular feature of one length scale with attributes of other length scales. 

Due to the diversity and inherent variability of cuticular structures, naming these structures has historically been complex. To date, there has been no comprehensive or uniform ontology established relating to insect surface cuticular morphologies. A recent review [28] of the nano- and microstructures of insects has led to the creation of seven distinct categories: (1) simple nanostructures (dome-like or pillars), (2) simple microstructures (dome-like or pillars), (3) complex geometric nanostructures (varied shapes), (4) complex geometric microstructures (varied shapes), (5) scales (flattened hairs or setae), (6) hairs/setae (columnar structures, longer than wide) and (7) hierarchical structuring (any combination of the prior six categories). Another review [48] of the nano- and microstructures of plant and insect surfaces uses terms such as “setae”, “denticle”, “fractal”, “hemispheres”, “pillars”, and “layered cuticle”. However, a third and fourth review use terms such as “nipple-like protuberances”, “cloth-like microstructures”, “hairs”, “scales”, “nanopillars”, “nanostructures”, and “waxes” [21,49]. It is easy to see how these structures may be confused or overlooked by those seeking to draw inspiration from insects to design novel multifunctional surfaces. 

Although many terms have been used interchangeably, there are nuanced distinctions in some cases that may be important to note. For example, the terms “hairs”, “sensilla”, “setae”, “microtrichia”, “bristles”, and “spines” have been used interchangeably, and while they all promote hydrophobicity, they do not all have the same physiological function. Some hairs, such as setae, are often classified as sensilla because they have mechanosensory functions, while others, such as spines, do not have sensory cells (but can have sensory setae on them) [50]. These distinctions are often important to note when discussing multifunctionality. There also exists conflict between morphological and taxonomic terminologies. Here, for simplicity, we have adopted certain terms to reflect the physical appearance of the structures only (Table 1). Still, in creating an extensive and more detailed review of the literature, we will use the terms as they are used by the authors.

### 2.2. Literature Review Methodology 

Using a combination of the terms “wettability”, “hydrophobic”, “hydrophobicity”, “insects”, “arthropods”, “hierarchical”, “structure”, “biological”, “materials”, “nanostructure”, “microstructure”, “cuticle”, “exoskeleton”, and “integument”, the following four search strings were created to search titles, abstracts, keywords, or all fields for relevant literature: (1) (wettability or hydrophob *) and (insects or arthropods) and (hierarchical and structure); (2) (wettability or hydrophob *) and (insects or arthropods) and (biological and materials); (3) (wettability or hydrophob *) and (insects or arthropods) and (nanostructure or microstructure); and (4) (wettability or hydrophob *) and (insects or arthropods) and (cuticle or integument or exoskeleton). These search strings were entered into the Scopus and Web of Science (Core Collection) databases. In May 2022, Scopus returned 341 results and Web of Science returned 290 results. Of the combined results returned by the two databases, 112 total references were determined to be relevant for further review. Criteria for relevancy was based on the discussion of cuticular structures and surface chemistry explicitly involved in the hydrophobic nature of insect cuticles. 

## 3. Hydrophobic Cuticular Structures Found in Insects

### 3.1. Nanoscale Hydrophobic Structures of Insects

Insects that solely exhibit nanostructures to induce hydrophobicity include some members of the orders Hemiptera, Odonata, and Ephemeroptera. These include cicadas and leafhoppers, dragonflies and damselflies, and mayflies, respectively. The wings of cicadas (Hemiptera: Cicadidae) have been shown to possess highly ordered nanopillar arrays that result in superhydrophobicity (Figure 3c,l, Table 2). These nanopillars are usually described as having a conical appearance with spherical caps, are arranged in a hexagonally packed pattern, and typically range from 100 nm to 500 nm in height. 

In the case of the black cicada (*Gudanga* sp. nr adamsi) the wings displayed both transparent and blackened regions. The transparent region followed the typical organized layout of nanopillars found in other cicadas but the nanopillars within the black region were found to be less ordered with diamond-shaped structures much larger than the structures of the transparent region [81]. This difference in structure is thought to be important regarding antireflective properties, an additional functionality of nanopillars [108]. 

The surface chemicals of *N. pruinosus* revealed by laser-ablation electrospray ionization imaging mass spectrometry (LAESI-IMS) included hydrocarbons, lipids, esters, amines, amides, and (un)sulfonated compounds [84]. An additional examination of *N. pruinosus* and *Magicicada cassinii* via gas chromatography-mass spectrometry (GC-MS) indicated the presence of short-chain fatty acids and saturated hydrocarbons ranging from C_17_ to C_44_ [85]. The same researchers also tried to tease apart the relative importance of nanopillar structure and surface chemistry. They showed that chemical extraction initially changes the shape of the nanopillars due to aspect ratio (height vs. width) changes, and functionality is compromised. However, when more of the outer layer is extracted, nanopillars are overall shorter and thinner, but again retain their pillar shape, and the functionalities (hydrophobicity and antibacterial activity) are recovered [85]. 

Rather than using integrated surface structures, leafhoppers (Hemiptera: Cicadellidae), employ the use of extra-cuticular particles known as brochosomes to induce superhydrophobicity (Figure 3d). These lipid-protein granules range in geometry (primarily hollow spherical dodecahedrons or truncated icosahedrons) and in size from about 200 nm to 1 μm [90]. They are secreted from specialized Malpighian tubules, spread on the body by the hind legs, and form a loose powdery coating the cuticle [89,90]. In some species, females produce highly interspecific cylindrical shaped brochosomes to coat their egg nests [109]. According to Rakitov, the synthesis and spreading of brochosomes is present in all major subfamilies of Cicadellidae [110].

Dragonflies and damselflies have also been shown to have nanopillars but rather than being well-ordered, they are usually disordered and not as uniform in size as the nanopillars found on cicada wings (Figure 3b,k). Dragonflies in the families Aeshnidae, Corduliidae, Libellulidae, and Gomphidae have all been reported to have randomly ordered nanopillars resulting in hydrophobic or superhydrophobic capability (Table 2). In addition to the nanoscale physical structures, it has also been shown that the epicuticular surface chemistry of dragonfly wings contribute to their hydrophobic nature (Table 2). Investigations of epicuticular waxes revealed the presence of long-chain aliphatic hydrocarbons, fatty acids, palmitic acids, alcohols, and esters [48,51,101,103,104]]. Nguyen et al. [101] reported the primary presence of *n*-alkanes with even-number chain lengths between C_19_ and C_26_ within the epicuticular lipids of the tau emerald dragonfly (*Hemicordulia tau*). Subsequently, Nguyen et al. [51] reported the presence of hydroxyls, alkyl hydrocarbons, ester carbonyls, and amide groups in the wandering percher dragonfly (*Diplacodes bipunctata*) and the black-faced percher dragonfly (*Diplacodes melanopsis*). 

Damselflies in the families Coenagrionidae, Calopterygidae, and Lestidae have been also reported to have physical structures appearing as randomly oriented nanopillars, wax rods, or oblate-shaped nanofibrils (Figure 3b,k, Table 2). 

Additionally, mayflies have been shown to have hydrophobic and superhydrophobic wing features. Wagner et al. [55] observed the wing topography of the blue-winged olive fly (*Ephemerella ignita*), the burrowing mayfly (*Ephoron virgo*), and *Ephemera vulgata*, finding them to be covered in a disordered array of nanopillars, much like the wings of Odonates, which induce a hydrophobic state. Byun et al. [54] reported the same fractal display of nanostructures in *Ephemera* sp. with a superhydrophobic contact angle of 153°.

### 3.2. Microscale Hydrophobic Structures of Insects

Insects that are hydrophobic due primarily to the microstructures on the surface of their exoskeleton include members of Coleoptera, Diptera, Hemiptera, and Hymenoptera. The microstructures may appear as hair-like features, scales, or wax structures (Figure 3b,h,k,o). While terrestrial insects are often concerned with repelling water or other potentially threatening liquids, aquatic insects may use their hydrophobic cuticular structures to trap thin air films against their bodies. Many aquatic insects employ these microstructures to create a plastron (i.e., gas gill, physical gill) which allows for gas exchange and enables them to breathe underwater for extended periods of time. There are two types of plastrons: compressible and incompressible. Compressible plastrons are bubbles of air that adhere to the surface of an insect but are unsupported and eventually collapse over time or at increasing water depths. Incompressible plastrons are supported by hydrophobic structures on the insect’s surface and the air volume will remain relatively the same for longer periods of time and despite water depth [87]. 

Terrestrial beetles with hydrophobic microstructures include darkling beetles (*Lagria hirta*, *Zophobas morio*), the flower chafer beetle (*Mimela testaceipes*), poplar leaf beetle (*Chrysomela populi*), terrestrial leaf beetle (*Gastrophysa viridula*), and the Namib desert beetle (*Onymacris unguicularis*). The cuticular features of these beetles resemble hair-like setae which promote hydrophobicity [1,54,55,58,59]. Examination of a terrestrial leaf beetle (*G. viridula*) revealed that microscopic structures on the tarsi allow the beetle to walk on substrates while under water, resulting in contact angles of roughly 110°. Setae on the tarsal segments (tarsomeres) trap air bubbles between the tarsomere and substrate, allowing for adherence and the ability to walk across surfaces while fully submerged [57]. The troutstream beetle (*Amphizoa sinica*) has microscale setae on its wings that result in hydrophobic properties [48]. Both the water lily leaf beetle (*Galerucella myphaea*) and the golden edge diving beetle (*Cybister chinensis*) are aquatic beetles that are capable of plastron respiration. The water lily leaf beetle has uniformly oriented setae on its elytra that curve toward the posterior region. Single water droplet testing revealed these structures to be superhydrophobic and capable of maintaining an air film for up to two days [58]. The elytra of golden edge diving beetles exhibit varying arrangements of polygonal surface sculpturing, pores, and channels, resulting in a wettability gradient decreasing from anterior to posterior regions [60]. The study also revealed functional chemical groups such as alicyclic alcohols, carbonyls, amides, and unsaturated hydrocarbons, as well as, both physical and chemical differences between fresh and dry samples, and female and male samples [60].

Dipterans with microscale features on their cuticular surface include March flies (Bibionidae), thick-headed flies (Conopidae), soldier flies (Stratiomyidae), the alkali fly (*Ephydra hians*), and the intertidal midge (*Clunio marinus*). Sànchez-Monge et al. [67] reported the presence of microtrichia on the wings of members belonging to Bibionidae, Conopidae, and Stratiomyidae which were characterized as hydrophobic. Alkali flies have hair-like structures covering their entire body along with a hydrocarbon-rich cuticle surface (primarily straight-chain alkanes) that induce a superhydrophobic state as they dive underwater to lay their eggs [72]. Like the alkali fly, the cuticle of the intertidal midge is also covered in a dense layer of microtrichia. The microtrichia contain both epicuticular lipids and a protein matrix. When submerged underwater the microtrichia demonstrated the ability to create a thin air film around the insect’s body and a large air bubble under the ventral abdomen between the legs with an estimated contact angle of 140° [73]. 

Hemipterans that rely on microstructures to repel water include water-treaders (*Mesovelia* spp.), saucer bugs (*Ilyocoris cimicoides*), lesser water boatman (*Corixa punctata*), and other corixid bugs (e.g., *Agraptocorixa eurynome*) which are covered in hair-like structures, and the poplar spiral gall aphid (*Pemphigus spyrothecae*) which uses waxes to protect itself from its sticky exudates inside the gall. Flynn and Bush [88] discuss the two-tiered microtrichia of water-treaders in their study on plastron respiration mechanisms involving arthropods and spiders. *Mesovelia* have hydrophobic structures both on their legs which allow them to walk on the water’s surface and have hairs on their abdomen which can support a plastron for short periods of time. The plastron surrounds their spiracles and enables them to exchange gas in case of submergence [88]. The superhydrophobic cuticular hairs of the saucer bug can support two types of plastrons; one involving setae on the abdominal sternites that can maintain an air film for two days and another involving microtrichia on the elytra that can maintain an air film for greater than four months [58]. Balmert et al. [58] also reported the presence of two types of setae on the abdominal sternites of lesser water boatman that could maintain an air film for two days. In a discussion on the physics of bubble gas exchange in collapsible plastrons, Seymour and Matthews [87] note the presence of collapsible gas gills on the ventral surface and hemelytra of *A. eurynome*. This capability is most likely due to hair-like structures found in other related corixid insects. 

Transitioning to terrestrial hemipterans, the poplar spiral gall aphid has a very distinct method of staying clean and dry. Insects living inside or causing damage to plant tissues commonly elicit a growth response within the plant that creates abnormal growths called galls. These galls can offer protection for developing insects but can also create issues due to the confined interior space of the gall. The poplar spiral gall aphid runs into this exact problem. As sap-feeding insects, aphids produce a large amount of liquid excrement, also known as honeydew. It’s easy to see how this would become a problem inside a constricted gall. To escape their liquid entrapment, the aphids secrete a powdery, needle-like wax, consisting of long-chain esters, which coats the inside of the gall and turns any liquid drops of honeydew into superhydrophobic “liquid marbles” that can then be maneuvered around and transported out of the gall [91].

Hymenopterans, such as pollen-collecting bees which are covered in dense wettable hairs, are not generally known for having hydrophobic attributes, but studies have revealed hydrophobic microstructures on the bodies’ surfaces in European honey bees (*Apis mellifera*), German wasps (*Paravespula germanica*), red wasps (*Vespula rufa schrenckii*), lesser paper wasps (*Parapoly varia*), flower wasps (*Scolia soror*), and elm sawflies (*Arge captiva*) [54,55,81,94]. In 1996, Wagner et al. [55] demonstrated that the fore- and hindwings of European honey bees are coated with short, thick hairs and the fore- and hindwings of German wasps are coated with longer, longitudinally twisted hairs both of which enable hydrophobicity. In a review of the wetting characteristics of insect wings, Byun et al. [54] attributed the wing hydrophobicity of red wasps, lesser paper wasps, and elm sawflies to the distribution of fine hair-like structures, either slightly curved at the distal end or curved evenly along the entire hair on the wing. Hu et al. [81] reported the dome-like nanostructures of *S. soror* wings to have a hydrophobic contact angle. A more recent study determined that honey bees tongues are coated in stiff superhydrophobic hairs, a surprising finding being that their tongues are adapted to collect floral nectars and other liquids such as aqueous saps, plant juices, and water [33]. Although the hairs are superhydrophobic, they provide structural integrity to promote flexibility of the tongue and facilitate the movement of liquids from the distal segment of the tongue hair to the proximal segment where the hair is attached to the base of the tongue. 

### 3.3. Macroscale Hydrophobic Structures of Insects

Although hydrophobic macroscale structures exist, they have not been reported as the sole method of hydrophobicity in insects, but instead are accompanied by nanoscale or microscale architecture [21,35,41,48,51,52,56,81,100,107]. Moreover, they are consistently arranged within beds of nano- or microstructures or they themselves are hierarchical structures with nano or micro topography. In theory, macrostructures could repel water droplets on single points in the Wenzel state with sufficient surface chemistry. Macrostructures could also potentially support water droplets in the Cassie–Baxter state if the features are adequately layered or in close enough proximity to one another to support an air film between larger droplets and the underlying surface of the cuticle. 

### 3.4. Hierarchical Hydrophobic Structures of Insects

Hierarchical systems are by far the most common strategy to reduce wettability in insects. This includes the use of structures that are independently hierarchical (i.e., where each structure of one length scale has attributes of at least one other length scale) or structures in hierarchical arrangements (i.e., where there are multiple structures of varying length scales in a given area). The taxonomic orders Blattodea, Coleoptera, Collembola, Diptera, Hemiptera, Hymenoptera, Lepidoptera, Mecoptera, Megaloptera, Odonata, Orthoptera, Neuroptera, and Trichoptera all contain insects with hierarchical structures. 

#### 3.4.1. Blattodea 

Within Blattodea, there are a few species of termites with hierarchical hydrophobic structures [52,111]. The wings of tree termites (*Nasutitermes walkeri*) and *Microcerotermes* sp. were found to have both hierarchical structures and hierarchical arrangements of structures [28] The wing membrane surface is covered with evenly spaced hairs and micrasters (i.e., star-shaped microscale protuberances). The hairs, characterized as macrotrichia, were found to contain nanogrooves along their longitudinal axis and the surface of the micrasters displayed open radial sheet-like folds [52]. Using a thick PDMS coating to smooth out the nanogrooves on the hairs, the authors were able to demonstrate that the nanoscale troughs greatly improved the hydrophobicity of the hairs. 

#### 3.4.2. Coleoptera

Several species of beetles (Coleoptera) display hierarchical structures that induce hydrophobicity (Figure 3a–c,e,f,h,i,k,o). Perhaps the most famous example is the Namib desert beetle (*Stenocara* sp.) which has alternating rows of non-waxy, hydrophilic micro-domes and waxy, hydrophobic textured troughs that, paired with behavioral maneuvers, are used to capture water from fog layers in the desert [28,41,56]. Hinton [35] studied the biology and structure of plastron respiration in 32 species of Hemiptera and Coleoptera including the families Elmidae, Chrysomelidae, and Curculionidae (Table 2). Each of these species were reported to have a mixture of hierarchical structures including scales or sternites with high geometric variability, and hairs, all capable of creating air plastrons [35]. Hinton’s [35] extensive scanning electron micrographs revealed scales and sternites covered with pits (either centered or distributed across the scales), ridges and troughs, longitudinal or radial folding, wrinkling, nanopillars, micro-domes, and hairs, some of which were sharp and pointed, leaf-like, branch-like, or serrated. The edges of the scales were either smooth or with projections that were fringed or finger-like and the overall scale shape could be rounded, pentagonal, hexagonal, or oblate [35]. Sun et al. [60] used scanning electron microscopy, white light interferometry, contact angle measurements, and chemical treatments to determine that the epicuticle of the dung beetle, *Geotrupes stercorarius*, is hydrophobic. The elytra are covered with a cement layer and wax particles where both the roughness and surface chemistry of the cuticle are sufficient to impart hydrophobicity. In the case of the polyphemus beetle (*Mecynorhina polyphemus confluens*) we can see that sometimes sexual dimorphism leads to differences in wettability. On females, water readily spreads across the elytra indicating hydrophilicity, while on males, water does not readily spread across the elytra. Closer examination revealed that the tomentose (i.e., dense hair covered) portion of the female elytra was covered in disorganized nanoscale needle-like structures separated in bunches by cracks across the surface. Meanwhile, the male elytra were found to be covered with uniform, vertically aligned, microscale and nanoscale needle-like structures [61]. Recently, a research team in Australia observed an unidentified beetle (most likely family Hydrophilidae) that was capable of walking upside-down on the underside of the water’s surface without penetrating the surface. The appearance of an air bubble on the abdomen of the beetle was visible but currently, the exact mechanisms of how the beetle is capable of this feat or what cuticular features are involved are unknown [112].

#### 3.4.3. Collembola

A fascinating display of complex geometric structures can be seen in several species of Collembola (i.e., springtails). The cuticle structure is often described as having a multiscale hexagonal, rhombic, or a comb-like appearance with overhanging re-entrant features. (Figure 3m). In *Orchesella cincta*, the cuticle surface was reported to have microscopic hairs (with feather-like geometry) protruding from the surface and nanoscopic ring-shaped nanocavities consisting of overhanging, primary, mushroom-shaped granules interconnected by ridges [63]. These complex hierarchical structures together with rich aliphatic hydrocarbons, glycine-rich structural proteins, fatty acids, fatty and sterol esters, terpenes, steroids, and triglycerides make the cuticle of *O. cinta* superhydrophobic [62]. Other springtails were found to have similar hierarchical features (Table 2). 

#### 3.4.4. Diptera

Many members of the order Diptera (i.e., flies) have hydrophobic hierarchical structures on their wings, eyes, abdominal segments, and legs (Figure 3c,h,o). The tiger crane fly (*Nephrotoma australasiae*) was found to have two distinct types of microscopic hairs on its wings and four types of microscopic hairs, some with nanogrooves, on its legs [66]. Wings of numerous other flies (Table 2) were also found to exhibit various nanoscopic and microscopic hairs, some with nanogrooves, on their wings [1,54,55]. The eyes of mosquitos and green bottle flies have also been studied for their superhydrophobic properties. The compound eye of the northern house mosquito (*Culex pipiens*) is composed of hexagonally close-packed microscale ommatidia that are individually coated with hexagonally non-close-packed nanopillars (Figure 3c) [69]. The compound eye of the green bottle fly (*Lucilia sericata*) is arranged in a similar manner but with close-packed nanopillars on each ommatidia [70]. As larvae, mosquitos are aquatic and use a snorkel-like breathing siphon at the distal tip of their abdomen to breathe. To keep their inner respiratory system dry and the snorkel on top of water while taking in air, coastal rock pool mosquito larvae (*Aedes togoi*) have hydrophobic lobes, (three main lobes and two auxiliary lobes) surrounding the siphon [71]. The microscale hydrofuge lobes open around the siphon on top of the water and are closed by hydrostatic pressure when submerged. Although nanoscale measurements were not directly reported, the presence of submicron hairs and surface sculpturing can be seen in the scanning electron micrographs provided by Lee et al. [71]. The presence of lipids, indicated by the use of Nile red, a fluorescent hydrophobic probe, was also reported. It is believed that the lobes are capable of secreting oily mixtures like the lobes of *Anopheles* and *Culex* mosquito larvae which are thought to aid in hydrophobicity [113].

#### 3.4.5. Hemiptera

Hemipterans (i.e., true bugs) are also known for their multilevel hydrophobic structures (Figure 3a–c,e,f,h,k,o). The wings of the planthopper (*Desudaba danae*) have cuticular structures that are analogous to the lotus leaf [107]. The hindwing surface is covered in a uniform widespread distribution of microscale domes and nanopillars. The forewing also displays a two-tiered topography with the microscale projections being clumped raised regions of nanopillars. Unsurprisingly, several aquatic hemipterans were found to have hierarchical structures as well. Plastron structures (scales, sternites, and hairs) can be seen in numerous creeping water bugs, saucer bugs, backswimmers, and pond skaters (Table 2) [40,58,87,88]. Other hierarchical cuticular arrangements have been found in a variety of Hemipterans (Table 2). These structures include multiple variations of nanopillars, hairs such as microtrichia and setae, either sharp and pointed, serrated or leaf-like (some with nanogrooves or nanopillars), and micro-domes [35].

#### 3.4.6. Hymenoptera

Hymenopterans with multiscale cuticular structures include sawfly larvae (*Rhadinoceraea micans*), wasps (*Vespa* sp.), yellow hornets (*Vespa simillima xanthoptera*), black hornets (*Vespa dybowskii*), fire ants (*Solenopsis invicta*), and the hind legs of male fig wasps (*Ceratosolen corneri*) (Figure 3b,c,e,h,k,o, Table 2). Sawfly larvae have complex nanoscale geometric sculpturing grouped into larger microscale domes along with microscale wax crystals that resulted in hydrophobicity [92,93]. The wings of *Vespa* species have nano- and microscale hairs, some with additional nanogrooves [48,54]. A distinct case of hydrophobicity can be seen in fire ant rafting behavior during flood periods. Single fire ants are only moderately hydrophobic but when linked together, their hydrophobicity increases by 30%, raising the contact angle from 102 ± 4° to 133 ± 12° [96]. The cuticle of a fire ant is covered in microscale hairs but when linked mandible to tarsus or tarsus to tarsus the ants can take advantage of the Cassie–Baxter state at the micro- and macroscale level. By tightening or loosening their grip to form rafts with their bodies, the ants are able to trap a plastron layer of air around themselves to stay dry and resist submersion [96]. In polyphemus beetles, we saw that sexual dimorphism in cuticular topography sometimes influences hydrophobicity, with males being more hydrophobic. The same is true of the male fig wasp, *Ceratosolen corneri*. Rodriguez et al. [95] found that the highly modified hind legs of these male fig wasps allow them to access submerged females earlier than male fig wasps without modified legs (*Ceratosolen bisulcatus*). Scanning electron microscopy revealed that the hind legs of *C. corneri* are covered with a sparse coating of nanopillars, and denser coatings of microtrichia and setae (some setae with raised bases) and modified underlying cuticle [95].

#### 3.4.7. Lepidoptera

Members of the order Lepidoptera (i.e., butterflies and moths) are often recognized for the photonic structures on their wings that cause striking optical effects but are rarely noted for other functional characteristics, such as hydrophobicity, induced by these same structures [54]. Morpho butterflies (*Morpho didius*, *Morpho Menelaus*, *Morpho aega*, etc.) are a prominent example of this phenomenon. The wings are comprised of an arrangement of aligned ground scales and cover scales overlayed with hierarchical microgrooves and nanostructures consisting of cross-ribs separated by longitudinal ridges (Figure 3j,p) [35,98]. These structures create brilliant iridescent blue coloration but also make the wings superhydrophobic. Other Lepidopterans have similar superhydrophobic wing arrays (Table 2). As a caterpillar, the ground lackey moth (*Malacosoma castrensis*) is hydrophobic as well [97]. The cuticle of the caterpillar is covered in microtrichia and setae which enable it to form a compressible plastron of air around its body when submerged in tidal zones up to 8 h, twice a day [97].

#### 3.4.8. Orthoptera

Other insects with hierarchical architectures include crickets, grasshoppers, and locusts in the order Orthoptera (Figure 3c,e,f). The wings of the sickle-bearing bush cricket (*Phaneroptera falcata*), Chinese grasshopper (*Acrida cinerea cinerea*), long-headed grasshopper (*Atractomorpha latta*), common field grasshopper (*Chorthippus brunneus*), mottled grasshopper (*Myrmeleotettix maculatus*), and the oriental migratory locust (*Locusta migratoria*) have nano and micro hydrophobic structures. The Chinese grasshopper and long-headed grasshopper are superhydrophobic, having nanoscale hairs on micro-domes across the wings (Figure 3f) [48,54]. The remaining Orthopterans have micro-domes or tooth-like protuberances, both with nanogrooves [55].

#### 3.4.9. Neuroptera

The wing membranes and veins of various Neuropterans (Table 2) were described to have nano, micro, and macro cuticular features resembling disorganized arrays of nanopillars and hairs (Figure 3b,h,i,k,o). A lattice of veins stretches across the wings and are covered in bundles of macrotrichia angled toward the center of the wing membrane cells. The surface of each macrotrichia is covered in ridges along the longitudinal axis and additional nanogrooves can be seen on the microscale ridges and troughs. The contact angle of droplets resting on these microtrichia was measured to be roughly 180° [107]. The wing membrane was also observed to have nanopillar-like structures that create a dense netting, resulting in a contact angle higher than 150° [54,55,81,107].

#### 3.4.10. Mecoptera, Megaloptera, and Trichoptera

Scorpion flies (order Mecoptera), alderflies (order Megaloptera), and net-spinning caddisflies (order Trichoptera) have been reported to have hierarchical structures on their wings that promote hydrophobicity (Figure 3h,o) [55]. Scorpion flies (*Sialis lutaria*) have a dense coating of curved hairs on their wing similar to Dipterans, along with longer straight hairs in the medial region of the wing. Alderflies (*Panorpa vulgaris*) have two-tiered hairs, a dense coating of smaller hairs and longer hairs arranged in rows across the wing. The net-spinning caddisfly (*Hydropsyche pellucidula*) exhibit a dense coating of microtrichia with uniformly distributed setae roughly eight times longer and slightly flattened [55].

## 4. Discussion and Bioinspired Design Implications

Presently, there are a few examples of how insects have inspired new technologies. Multiscale copper hydroxide nanoneedle arrays with nanogrooves, inspired by water strider legs, have been fabricated on copper materials to create novel surfaces with superhydrophobic characteristics [68]. A bioinspired templating technique has been developed to fabricate multifunctional optical coatings based on the superhydrophobic self-cleaning nanopillars of cicada wings and the anti-reflective compound eyes of moths [68]. Another novel templating technique, dissolvable template nanoimprint lithography (DT-NIL), was created to replicate the nanopillar structure of cicada wings [83]. Other soft lithography techniques have allowed us to create materials inspired by the superhydrophobic and anti-fogging properties of mosquito compound eyes and superhydrophilic surfaces, inspired by the anti-reflective and anti-fogging properties of insects, have been created using silica materials [68]. Based on principles derived from the elytra of the Namib desert beetle, one research group used theoretical modeling to create a surface with a sixfold-higher exponent growth rate for condensing and collecting water [114]. Nowlin and LaJeunesse [115] demonstrated that modifying the nanosphere lithographic (NSL) technique by using different substrates, altering etching techniques, or reiterating the nanosphere lithographic process itself can result in the production of hydrophobic biomimetic surfaces that mimic nanoscale hierarchies found on insect cuticles.

Inspired by the hierarchical structures of nanoscale interconnected granules with re-entrant curvatures found on the cuticle of springtails, Agonafer et al. [116] developed a novel approach for retaining low-surface-tension liquids behind a porous membrane on a silicon surface. Their liquid retention strategy can facilitate the routing and phase management of dielectric work fluids in heat exchangers of electronic systems and has further applications in oil transportation, water/oil separation, microfluidics, and thermal managements of power systems [116]. By etching dragonfly-inspired nanopillars onto black silicon (bSi), scientists were able to create a microfluidic flow channel that successfully killed 99% of *Pseudomonas aeruginosa* and *Escherichia coli* bacteria in water [116,117].

As we begin to uncover and understand how hydrophobic mechanisms of insects function, we tap into a relatively unstudied wealth of resources. Hydrophobic cuticular features have incredible potential in almost all categories of engineering and material science applications. Since hydrophobicity frequently elicits antimicrobial behavior and other functionalities, one major prospective application of insect-inspired surfaces is within the medical field [28,35,48,53,75,118]. The efficacy of surgical tools, biomedical implants, prosthetics, medical devices, and medical diagnostics tools could all potentially be improved by implementing functional attributes of various insect cuticle structures. Nguyen [48] suggests that rough Cassie–Baxter structures hold the most promise for synthetic superhydrophobic self-cleaning applications. Insect-inspired patterning of medical devices has the potential to reduce microbial growth, enhance tissue scaffolds, or act as a substrate for in vitro tissue regeneration [28].

Respiratory-related cells and organ-on-a-chip systems could benefit from insect-plastron-like designs for nutrient and gas exchange [1]. The formation of biofilms could also be prevented from occurring by using cicada-inspired nanopillar textures, which have been shown to inhibit subsequent infections of the surrounding tissue [53]. Webb et al. [77] provided extensive examples of insect-inspired medical applications, including superhydrophilic micro-patterned platforms that can control fibroblast adhesion, hemocompatible implants that inhibit blood platelet coagulation and minimize unwanted immune response (i.e., medical implant rejection), and superhydrophobic surfaces that contain microdroplets for microfluidic devices and targeted drug delivery systems, etc.

Other applications for insect-inspired surfaces include protective and self-cleaning paints and coatings for vehicles and buildings, hydrophobic antennas, windows, windshields of vehicles, non-medical microfluidics devices (e.g., no-loss analysis channels), metal surface refinements for applications in energy systems and computing components, and hydrophobic antimicrobial textiles [36,118,119,120]. Furthermore, (super)hydrophobic materials paired with other attributes, such as structural color or transparency, seen in the *Hoplia coerulea* beetle, could be used for specialized self-cleaning coatings on solar cells and panels [49]. Importantly, major limiting factors of manufacturing insect-inspired surfaces include the fact that current fabrication techniques are expensive, technically challenging, require sterile environments and state-of-the-art machinery, and are commonly incapable of high throughput [53]. Nevertheless, nano- and microfabrication techniques are improving.

## 5. Outlook

Organisms are constantly adapting to environmental stress and variation. The insect cuticle, serving as a barrier between the insect and its external environment, is exceedingly variable and capable of remarkable functionalities, including adhesion, chemical sensing and defense, color manipulation, locomotion, mechanosensation, thermoregulation, (anti)reflectivity, and (super)hydrophobicity [1]. This multifunctionality is achieved through a wide variety of surface chemistries and insect cuticular surface structures.

Recent research developments have revealed compelling functional properties of over two hundred insects, and yet we still only know a small fraction of all the cuticular characteristics that millions of described insect species have to offer. While the insect cuticle hosts a wide range of functional attributes with applications ranging from locomotion to photonics, wettability characteristics are of particular interest as they often result in other desired functionalities such as anti-icing, anti-fogging, anti-corrosion, fluidics control, self-cleaning behaviors, and antimicrobial activity [28].

Using the bioinspiration framework, we can discover unique solutions in nature, analyze their capacity for functionalization, and, more importantly, their limitations, to create optimized designs and materials for our own societal and technological needs. However, we should proceed with caution. While there is a rushed tendency for biomimicry and replication, it is essential to understand how and why functional materials work in nature to determine which physical and chemical principles result in the functionalities observed. For example, the nanopillars found on cicada and dragonfly wings display antibacterial activity, but mimicking the exact topography might not be enough to kill bacteria if there are other factors at play, such as surface chemistry or involvement of the insect’s immune response [53,85]. Conditions in nature can also be more complicated than experimental settings. In nature, rapid rainfall frequently shatters on biological hydrophobic surfaces [121]. However, this is often overlooked when we test hydrophobicity in the lab using low-speed impact droplets. Using sessile or immobile droplets to test wettability on the wings of *Nasutitermes* sp. and *Microcerotermes* sp. termites did in fact reveal that the wings are superhydrophobic, but did not reveal that raindrops shatter and fragment away from the wings during colonization flights in rainy periods [111].

Historically, based on the Cassie–Baxter theory, low solid fraction textures were deemed essential in the creation of water-repellent materials. However, a recent study found that certain insects with *high* solid fraction textures are still able to achieve water repellency by reducing the texture size below 300 nm [122]. Additionally, this study postulated that the compact and nanoscale features of insect surfaces work favorably to rapidly shed high-impact water droplets such as rain [122], and thus maybe even carry away contaminating particles from a surface [123]. Passive self-cleaning has been shown to occur in nature through coalescence-induced jumping of microscale condensate droplets during fogging, dewing, or condensation, including in insects [79], even when just one droplet coalesces with a particulate of a certain size under those conditions [124]. Particle transport physics was shown to depend on the wettability characteristics of surfaces, including natural surfaces, e.g., butterfly wings, cicada wings, and clover leaves [124]. Studies such as these on rapid droplet shedding and droplet jumping, including particle-droplet jumping, can provide guidelines for the design of artificial water-repellent and passive self-cleaning surfaces.

Here, we have presented an extensive review of hydrophobic and superhydrophobic structures in insects. These properties are currently in high demand due to their associated functionalities. Insects across fourteen taxonomic orders were identified to possess a variety of cuticular surface chemicals and physical structures that promote hydrophobicity. Both terrestrial and (semi-)aquatic insects encounter water and other liquids that pose a variety of threats to the insect. The cuticle of insects serves as a barrier and can be highly modified to help the insect stay dry, moist and clean, keep vision clear, maintain flight, or enable underwater respiration and locomotion. Chemical compositions that enable hydrophobicity include long-chain aliphatic hydrocarbons, fatty acids, palmitic acids, alcohols, and esters [48,51,84,102,103,104], lipids, amines, amides, (un)sulfonated compounds [84], alicyclic alcohols, carbonyls, and unsaturated hydrocarbons [60]. Physical structures that enable hydrophobicity are widely variable in shape and can be part of the cuticle or rest on the cuticle surface. Physical protuberances can be in highly ordered or disordered arrays, with complex or simple geometry, range in size from nano to macro, and often make use of multi-level structuring and/or hierarchical arrangements [28].

Despite decades of research and an increasing interest in the wettability mechanisms of insects, only a little over two hundred insects have been reviewed, and many hydrophobic attributes have not yet been discovered or entirely understood. With over one million described insect species and an estimated 5.5 million undescribed species [125], we have just barely begun to explore the true extent of hydrophobic cuticular structures found in insects. We believe that the diversity of novel hydrophobic structures in insects will only continue to grow along with their potential applications and our discovery of new species, especially in hyper-biodiverse regions of the world [75]. It is therefore crucial to preserve insect biodiversity, not only to benefit our own society and industries, but to combat the challenges of human-related defaunation and climate change.

## Figures and Tables

**Figure 1 insects-14-00042-f001:**
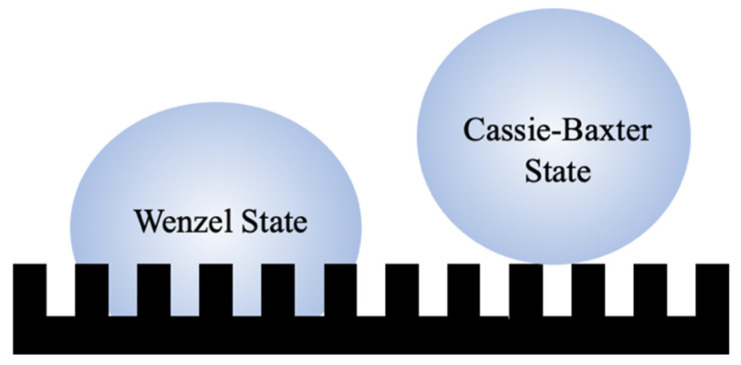
Schematic representation of water droplets suspended on a cuticular surface in the Wenzel state (**left**) and Cassie–Baxter state (**right**). Note: Droplets and texture beneath droplets are not drawn to scale.

**Figure 2 insects-14-00042-f002:**
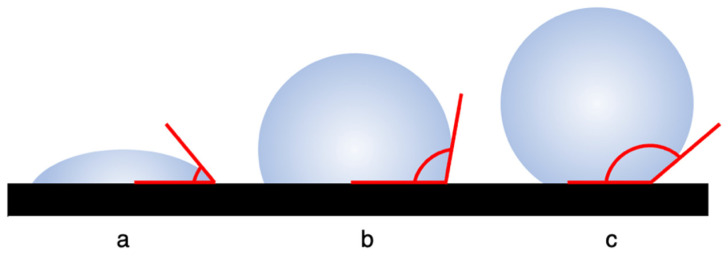
Schematic representation of wettability characterization: (**a**) hydrophilic contact angle (CA) less than 90°; (**b**) hydrophobic CA greater than 90°; (**c**) superhydrophobic CA greater than 150°. Note: This diagram is not designed to show surface roughness beneath the droplets.

**Figure 3 insects-14-00042-f003:**
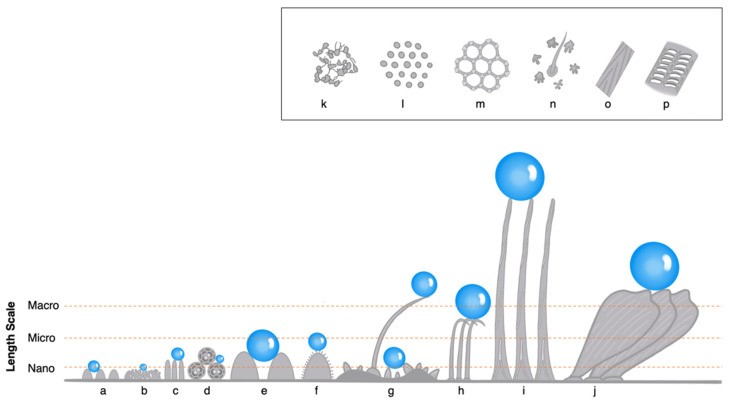
Main illustration: Side profile of select cuticular surface structures of insects (**a**) nano-domes (**b**) disordered nanopillars or wax structures (**c**) ordered nanopillars (**d**) brochosomes (**e**) micro-domes (**f**) hierarchical micro-domes with nanostructures (**g**) hierarchical arrangement of micrasters and hairs with nanogrooves (**h**) microscale hairs (**i**) macroscale hairs with nanogrooves (**j**) cover scales with nanogrooves. Inset: Top view of select cuticle surface structures of insects (**k**) disordered nanopillars or micro-domes (**l**) ordered nanopillars or micro-domes (**m**) polygonal cuticular patterning (**n**) hierarchical arrangement of micrasters and hairs with nanogrooves (**o**) micro- or macroscale hairs with longitudinal nano- or microgrooves (**p**) hierarchical microgrooves with nanoscale cross-ribs separated by longitudinal ridges. Note: cuticular features drawn roughly to length scale, water droplet size enhanced (not to scale) for visualization purposes.

**Table 1 insects-14-00042-t001:** Adopted terminology and synonyms used to describe cuticular structures.

Adopted Term	Synonym
Hierarchical	Multi-scale, multi-layer, layered
Cuticle	Exoskeleton, integument
Sculpturing	Textured, pattern, topography
Ordered	Homogenous, non-random, organized
Disordered	Inhomogeneous, nonhomogeneous, heterogeneous, random, disorganized
Pillar	Nipple, nipple-array, nipple-like, tapered rod, wax rod, wax needle, conical protrusion, conical protuberance, protrusion, protuberance, projection, denticle
Dome	Bump, hemisphere, protrusion, protuberance, projection, papillae, denticle, ridge
Hair	Setae, sensillum, trichia, bristle, spine
Wax	Wax crystals, wax rods, wax needles, wax particles, wax powder
Structure	Feature, sculpture
Groove	Cavity, trough
Pattern	Array
Scale	Sternite
Particle	Granule
Brochosome	N/A
Micraster	N/A
Ground Scale	N/A
Cover Scale	N/A
Cross-rib	N/A

**Table 2 insects-14-00042-t002:** Cuticular surface structures of insects.

Order	Insect	Structure	Length Scale	References
Blattodea	Tree termite (*Nasutitermes* sp.)	Macrotrichia with troughs Micrasters Hairs	Hierarchical (nano, micro, macro)	[51,52]
	Termite (*Microcerotermes* sp.)			
	Termite wings	Microsetae with nanogrooves Micrasters Nanobumps Microbumps Micropillars Denticles	Hierarchical (nano, micro)	[28,48,53]
	Tree termite (*Nasutitermes walkeri*)			
Coleoptera	Namib desert beetle (unidentified species)	Hydrophilic microbumps with hydrophobic troughs Wax Hexagonal array of flattened micro-hemispheres Setae Alternating hydrophobic and wax-coated, and hydrophilic and non-waxy, regions Macro near-random array of bumps Flattened hemispheres microstructure (regular hexagonal array)	Hierarchical (micro, macro)	[1,21,28,41,48,54,55,56]
	Desert toktokkie beetle (*Physaterna cribripes*)			
	Terrestrial leaf beetle (*Gastrophysa viridula*)	Setae		[57]
	Troutstream beetle (*Amphizoa sinica*)			[48]
	Darkling beetle (*Stenocara* sp.)			[54,58]
	Water lily leaf beetle (*Galerucela myphaea*)	Physical gill setae	Micro	[1,55,59]
	Sun beetle (*Pachnoda marginata*)	Setae		
	Darkling beetle (*Lagria hirta*)			
	Darkling beetle (*Zophobas morio*)			
	Flower chafer beetle (*Mimela testaceipes*)			
	Poplar leaf beetle (*Chrysomela populi*)			
	Aquatic beetle (*Stegoelmis* sp.)	Plastron scales Plastron sternites Plastron hairs	Hierarchical (nano, micro, macro)	[35]
	*Tyletelmis mila*			
	*Elsianus isus*			
	*Elmis maugei*			
	*Limnius volckmari*			
	*Austrolimnus waterhousei*			
	*Austrolimnus formosus*			
	Riffle beetle (*Hexacylloepus nunezi*)			
	Pilelmis halia			
	Riffle beetle (*Cylloepus barberi*)			
	Riffle beetle (*Cylloepus caicus*)			
	*Portlemis nevermanni*			
	*Lixellus haldemani*			
	Leaf beetle (*Macroplea mutica*)			
	Horsetail weevil (*Grypus equiseti*)			
	Weevil (*Bagous americanus*)			
	Weevil (*Bagous limosus*)			
	Golden edge diving beetle (*Cybister chinensis*)	Ordered submicron-scaled pits along polygonal edges and pores	Micro	[60]
	Dung beetle elytra (*Geotrupes stercorarius*)	Cuticle structure Cement Wax layer	Hierarchical (nano, micro)	
	Polyphemus beetle (*Mecynorhina polyphemus confluens*)	Vertically aligned needles (tomentose section on the elytra)		[61]
Collembola	*Orchesella cincta*	Multiscaled rough structure Nanocavities Surface chemistry: lipids with a diverse carbon number triacylglycerols Nanometer-thin triacylglycerol-containing wax layer at the cuticle surface Nanoscopic, comb-like structures	Hierarchical (nano, micro)	[62] [63]
	*Entomobrya intermedia*	Setae Hexagonal or rhombic comb-like patterns		[1,21,28]
	*Pogonognathellus flavescens*			
	*Vertagopus arboreus*			
	*Isotoma viridus*			
	*Kalaphorura burmeisteri*			
	*Stenaphoruella quadrispina*			
	*Dicyrtomina ornata*			
	*Arrhopalites pygmaeus*			
	*Tetrodontophora bielanensis*			
	*Tomocerus fivescens*	Hexagonal sculpturing Primary (small) granules with ridges connecting primary structures Secondary (large) granules		[37,64,65]
Diptera	Tiger crane fly (*Nephrotoma australasiae*)	Four hair types, some with nanogrooves	Hierarchical (nano, micro)	[66]
	March flies (*Bibionidae*)	Microtrichia	Micro	[67]
	Thick-headed flies (*Conopidae*)			
	Soldier flies (Stratiomyidae)			
	Mosquito compound eye	Hexagonally close-packed micro-ommatidia Hexagonally non-close-packed nano-nipples	Hierarchical (nano, micro)	[1,21,68]
	Northern house mosquito compound eye (*Culex pipiens*)	Microhemispheres (ommatidia) Hexagonally non-close-packed nanonipples on ommatidia		[69]
	Green bottle fly eye (*Lucilia sericata*)	Hexagonally close-packed nanonipples on ommatidia		[70]
	Marsh crane fly (*Tipula oleracea*)	Nanoscopic and microscopic hairs, some with nanogrooves, on their wings		[54] [1,55]
	Pale giant horse fly (*Tabanus bovinus*)			
	Marmalade hoverfly (*Episyrphus balteatus*)			
	Drone fly (*Eristalis tenax*)			
	Noon fly (*Mesembrina meridiana*)			
	Horse fly (*Tabanus chrysurus*			
	Coastal rock pool mosquito larvae (*Aedes togoi*)	Hydrofuge lobe (snorkel-like breathing apparatus) Oil secretions Lipids		[1,71]
	House fly (*Musca domestica*)	Microtrichia with nanoscale grooves	Hierarchical (nano, micro, macro)	[36]
	Alkali fly (*Ephydra hians*)	Setae	Micro	[72]
	Intertidal midge (*Clunio marinus*)	Microtrichia		[73]
Ephemeroptera	Blue-winged olive fly (*Ephemerella ignita*)	Fractal	Nano	[55]
	Burrowing mayfly (*Ephoron virgo*)			
	Mayfly (*Ephemera* sp.)			[54]
	*Ephemera vulgata*			[55]
Hemiptera	Water strider (*Aquarius paludum*)	Microtrichia Setae	Hierarchical (nano, micro)	[74]
	Cicada wings (*Megapomponia intermedia*)	Nanopillars Nanostructures in orderly manner Nipple-like protuberances (cone like base with spherical cap) Denticle Cuticular nanoarrays Nanodomes Disordered inhomogeneous surface Nanostructured conically shaped protrusions	Nano	[21,34,48,49,53,55,68,75,76,77]
	Milky cicada (*Ayuthia spectabile*)			
	*Claripennis aguila*			
	*Pomponia scitula*			
	*Meimuna conica*			
	*Meimuna durga*			
	*Aola bindusara*	Nanostructure (protrusion)		[21,34,48,49,53,55,68,75,76,77,78]
	*Meimuna mongolica*			
	*Platylomia radha*			
	*Dundubia vaginata*			
	*Dundubia rasingna*			
	*Meimuna opalifer*			[21,34,48,49,53,54,55,68,75,76,77]
	*Terpnosia vacua*			
	*Terpnosia jingpingensis*			
	*Cryptotympana atrata*			[21,34,48,49,53,55,68,75,76,77,78]
	Clear-wing cicada (*Psaltoda claripennis*)			[21,34,48,49,51,53,55,68,75,76,77,79]
	*Chremistica maculata*			
	*Meimuna microdon*			
	*Zamara smaragdina*			
	Grey cicada (*Cicadia orni*)	Nipple array Nanocone array		[17,21,34,48,49,53,55,68,75,76,77,80]
	*Tettigia orni*			
	*Leptopsalta bifurcata*	Nanostructure (protrusion)		[78]
	Wattle cicada (*Cicadetta oldfieldi*)	Hexagonally packed spherically capped conical protuberances Clear membrane: similar well-ordered structure size, shape, and periodicity Black membrane: less-ordered surface with individual diamond-shaped structures Relatively large-sized curved projections (bumps), flat (low in height) and spaced many hundreds of nanometers apart		[81]
	Black cicada (*Gudanga sp. nr adamsi*)			
	Bladder cicada (*Cystosoma schemltzi*)			
	Scissor grinder cicada (*Neotibicen pruinosus*)	Nanopillars		[82,83,84,85]
	Dog day cicada (*Neotibicen tibicen*)			[82]
	Bush cicada (*Megatibicen dorsatus*)			
	Pharaoh cicada (*Magicicada septendecium*)			
	Dward periodical cicada (*Magicicada cassinii*)			[85]
	Backswimmer (*Notonecta glauca*)	Physical gill setae and microtricia Large sparse setae Small dense microtrichia Sharp-tipped setae Tapered-rod protective wing covers	Hierarchical (nano, micro)	[1,58,86]
	Backwimmer (*Anisops* sp.)	Compressible gas gill		[40,87,88]
	Backswimmer (*Notonecta* sp.)	Setae (clubs and pins)		Flynn and Bush, 2008 Mail et al., 2018
	Common water strider (*Gerris remigis*)	Numerous oriented needle-sharped microsetae with elaborate nanogrooves Papillae Spindly microsetae with nanoscale grooves	Hierarchical (nano, micro, macro)	[39]
	Common pond skater (*Gerris lacustris*)	Physical gill microtrichia Setae		[1,58]
	Water strider leg	Microsetae with nanogrooves Micrasters Nanobumps Microbumps Micropillars Microhairs Microtrichia	Hierarchical (nano, micro)	[21,53,68]
	Water-treader (*Mesovelia* sp.)	Two-tiered hair layer Microtrichia	Micro	[88]
	Leafhoppers (Cicadellidae)	Brochosomes with truncated icosahedral geometry Chemistry: protein and lipids Honeycomb-shaped hexagonal and pentagonal structures with re-entrant curvatures	Nano	[1,21,28,89,90]
	Planthopper (*Desudaba danae*)	Microprojections Nano-protuberances Micropillars	Hierarchical (nano, micro)	[1] [14]
	River bug (*Aphelocheirus aestivalis*)	Physical gill cuticular hairs Plastron hairs		[1,35,87]
	Coreidae dock bug (*Coreus marginatus*)			[55]
	Pentatomidae gorse shield bug (*Piezodorus lituratus*)	Cuticular hairs		
	Pentatomidae stink bug (*Carbula putoni*)			[54]
	Lantern bug (*Limois emelianovi*)			
	Naurocoridae saucer bug (*Ilyocoris cimicoides*)	Setae	Micro	[58]
	Naurcoridae (*Idiocarus minor*)	Plastron hairs	Hierarchical (nano, micro, macro)	[35]
	Naurcoridae (*Cataractocoris marginiventris*)			
	Naurcoridae (*Heleocoris mexicanus*)			
	Naucoridae water bug (*Cryphocricos mexicanus*)	Leaf-like setae		
	Helotrephidae (*Neotrephes usingeri*)	Plastron sternites with protuberances		
	Lesser water boatman (*Corixa punctata*)	Setae Microtrichia	Micro	[58]
	*Agraptocorixa eurynome*	Compressible gas gill		[87]
	Poplar spiral gall aphid (*Pemphigus spyrothecae*)	Powdery wax (long chain esters) Wax needles		[91]
Hymenoptera	Flower wasp (*Scolia soror*)	Nanostructures Relatively large-sized curved projections (bumps), flat (low in height), and spaced many hundreds of nanometers apart	Nano	[81]
	Sawfly larvae (*Rhadinoceraea micans*)	Complex sculptures Wax crystals Hill-shaped sculptures with radial ridges and crater-like tips	Hierarchical (nano, micro)	[92,93]
	Pollinator eyes	Ommatidia cuticular geometry	Nano	[75]
	Wasp (*Vespa* sp.)	Setae	Hierarchical (nano, micro)	[48]
	Yellow hornet (*Vespa simillima xanthoptera*)			
	Black hornet (*Vespa dybowskii*)			[48]
	European honey bee tongue (*Apis Mellifera*)	Dense hairs	Micro	[55,94]
	German wasp (*Paravespula germanica*)			[55]
	Red wasp (*Vespula rufa schrenckii*)			[54]
	Lesser paper wasp (*Parapoly varia*)			
	Elm sawfly (*Arge captiva*)			
	Male pollinator fig wasp hind legs (*Ceratosolen corneri*)	Setae Microtrichia	Hierarchical (nano, micro)	[95]
	Red imported fire ant rafts (*Solenopsis invicta*)			[96]
Lepidoptera	Giant blue morpho (*Morpho didius*)	Scales with aligned micro-grooves	Hierarchical (micro, macro)	[34]
	Moth wing (*Prasinocyma albicosta*)	Scales with typical overlaying tile type arrangement	Hierarchical (nano, micro)	[81]
	Ground Lackey caterpillar (*Malacosoma castrensis*)	Hair (setae) Microtrichia		[97]
	Malabar tree nymph (*Idea malabarica*)	Complicated composition of nano- and microstructures Nanostructures of cross-ribs separated by longitudinal ridges		[98]
	Citrus swallowtail (*Papilio xuthus*)			[51]
	Dark green fritillary (*Speyeria aglaja*)	Cover scales Nano-/microfeatures Layered cuticle Ommatidia nanonipples		[21,28,48,53,54,55,68,77]
	Menelause blue morpho (*Morpho menelaus*)	Cover scales Nano-/microfeatures Layered cuticle Ommatidia nanonipples Ordered microstructure Ground scales Wing scales		[21,28,48,53,54,55,68,77,99]
	Indian cabbage white (*Artogeia canidia*)	Cover scales Nano-/microfeatures Layered cuticle Ommatidia nanonipples		[21,28,48,53,54,55,68,77]
	Aega morpho (*Morpho aega*)	Cover scales Nano-/microfeatures Layered cuticle		
	Horse-chestnut leaf miner (*Cameraria ohridella*)			
	*Boarmia ribeata*			
	Wood carpet moth (*Cidaria rivata*)			
	Chinese tussar moth (*Autographa pernyi*)			
	Small skipper (*Thymelicus sylvestris*)			
	Cabbage white (*Pieris brassicae*)			
	Small tortoiseshell butterfly (*Aglais urticae*)			
	Marbled white (*Melanargia galathea*)			
	Mulberry tiger moth (*Lemyra imparilis*)	Setae with nanogrooves Macrotrichia with longitudinal ridges and troughs	Hierarchical (nano, micro, macro)	[100]
Mecoptera	Meadow scorpionfly (*Panorpa vulgaris*)	Setae with nanogrooves Macrotrichia with longitudinal ridges and troughs	Hierarchical (nano, micro, macro)	[55]
Megaloptera	Alderfly (*Sialis lutaria*)	Setae with nanogrooves Macrotrichia with longitudinal ridges and troughs	Hierarchical (nano, micro, macro)	[55]
Odonata	Vagrant darter dragonfly (*Sympetrum vulgatum*)	Randomly oriented nanopillars at various scales	Nano	[34,48,53,54,55,77]
	Red-veined darter dragonfly (*Sympetrum fonscolombii*)	Waxy epicuticular layer [with] unique surface nanoarchitecture that consists of irregular arrays of nanoscale pillars		[101]
	Dragonfly wings	Clear C-H stretching bands, prevalence of methylene bands which indicate long-chain aliphatic hydrocarbons		[76]
	Yellow-striped flutterer dragonfly (*Rhyothemis phyllis chloe*)	Rod-like structures		[81]
	Spread-winged damselfly (*Lestes sponsa*)	Wax rods Nanostructures Nanopillar array		[17,102]
	Australian emperor dragonfly (*Hemianex papuensis*)	Randomly oriented nanopillars Fractal Nanostructures Nanospikes Nanomembrane surface appeared as a nanoscale mesh with rough spikes		[51]
	Blue-tailed damselfly (*Ischnura elegans*)			[55]
	Common skimmer dragonfly (*Orthetrum albistylum speciosum*)			[103,104]
	Wandering glider dragonfly (*Pantala flavescens*)	Waxy covering and geometric non-smooth structure-column papillae Long-chain hydrocarbons, fatty acids, alcohols, and esters		
	Tau emerald dragonfly (*Hemicordulia tau*)	Nanostructures Nanoscale pillars Epicuticular lipids (primarily aliphatic hydrocarbons, especially *n*-alkanes with even-number chain lengths between C19 and C26, and relatively small proportion of palmitic acid)		[101]
	Brown darner dragonfly (*Gynacantha dravida*)	Oblate-shaped (chitin) nano-fibrils		[105]
	Blue riverdamsel (*Pseudagrion microcephalum*)			
	Swamp flat-tail dragonfly (*Austrothemis nigrescens*)	Waxy epicuticular layer with unique surface nanoarchitecture that consists of irregular arrays of nanoscale pillars Clear C-H stretching bands, prevalence of methylene bands which indicate long-chain aliphatic hydrocarbons		[101]
	Epaulet skimmer dragonfly (*Orthetrum chrysostigma*)			
	Violet dropwing dragonfly (*Trithemis annulata*)			
	Lesser emperor dragonfly (*Anax parthenope*)			
	Emperor dragonfly (*Anax imperator*)			
	Green-eyed hook-tail dragonfly (*Onychogomphus forcipatus*)			
	Wandering percher dragonfly (*Diplacodes bipunctata*)	Nanopillars Cuticular waxes: hydroxyl, alkyl hydrocarbons, ester carbonyl, amide groups, long-chain aliphatic hydrocarbons		[48]
	Black-faced percher (*Diplacodes melanopsis*)			
	Common bluetail damselfly (*Ischnura heterosticta*)			
	Red and blue damselfly (*Xanthagrion erythroneurum*)			
	Banded demoiselle damselfly (*Calopteryx splendens*)	Long wax rods on wing veins Wax crystals of various shapes	Hierarchical (nano, micro)	[106,27]
Orthoptera	Chinese grasshopper (*Acrida cinerea cinerea*)	Denticle	Hierarchical (nano, micro)	[48,54]
	Long-headed grasshopper (*Atractomorpha latta*)			
	Sickle-bearing bush-cricket (*Phaneroptera falcata*)			[55]
	Common field grasshopper (*Chorthippus brunneus*)			
	Oriental migratory locust (*Locusta migratoria*)			
	Mottled grasshopper (*Myrmeleotettix maculatus*)			
Neuroptera	Banded lacewing (*Glenoleon pulchellus*)	Nanostructures Wing membrane: interconnected ridges forming a dense netting on the cuticle surface Vein regions: array of macrotrichia	Hierarchical (nano, micro, macro)	[48]
	Common green lacewing (*Chrysoperla carnea*)			[55]
	Ant lion (*Grocus bore*)			[54]
	Mantid fly (*Mantispa* sp.)			
	*Glenuroides japonicus*			
	Green lacewing (*Chrysopa oculata*)	Mactrotrichia on wing veins (angled toward cells) with longitudinal ridges and troughs along hair shaft Dense netting of nanopillars on wing membrane		[107]
	Mantid lacewing (*Ditaxis biseriata*)			
	Australian blue eyes lacewing (*Nymphes myrmeleonides*)			
Trichoptera	Net-spinning caddisfly (*Hydropsyche pellucidula*)	Mactrotrichia on wing veins (angled toward cells) with longitudinal ridges and troughs along hair shaft Dense netting of nanopillars on wing membrane	Hierarchical (nano, micro, macro)	[55]

## Data Availability

Not applicable.

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
