# Peer review of "Staying Dry and Clean: An Insect’s Guide to Hydrophobicity"

_insects, 2022, doi:10.3390/insects14010042_

Round 1

Reviewer 1 Report

There are some English errors. Some examples are commented in the attached pdf. 

It is better to refer the figures such as SEM images in the references with permissions. They help the understanding of readers. 

Wetting phenomena on structured surface is essential. More detail description will help the readers' understanding. The following article may be suitable as a reference for physical understanding of wettability. 

Hiroyuki Mayama, Wetting Phenomena on Structured Surfaces: Contact Angle, Pinning, Rolling and Bouncing, Biomimetics, Jenny Stanford Publishing https://doi.org/10.1201/9781003277170

Author Response

Dear Reviewer 1 

We sincerely thank you for providing insightful and constructive comments on our work. We believe that your suggestions have improved the quality and clarity of the manuscript. Based on all 3 reviewers' input we made changes to the review paper, including re-naming of headings, further explanation of wettability, discussion of droplet jumping, etc.

Comment 1: There are some English errors. Some examples are commented on in the attached pdf. 

Response to Comment 1: We thank the reviewer for taking the time to read the manuscript in great detail and catch the typos and grammatical errors. 

  • Line 17 - the whole manuscript was written in “American English”. When using that syntax the correct usage would be “scientific literature”, not “scientific literatures”. We did not make the suggested change. 
  • Line 351 - the typo was correct and the sentence now uses “height”.
  • Line 352 - the comma was added
  • Line 659 - “more dense” was replaced by “denser”. 

Comment 2: It is better to refer to the figures such as SEM images in the references with permissions. They help the understanding of readers. 

Response to Comment 2: The reason why we created Figure 4 (a graphical representation of structures described in previous work) was to make it easier for the reader to compare the different structures. Adding actual images from previous work, with permission, instead of adding a graphic like Figure 4, would make the review a lot longer. We also hope that by adding a reference column to Table 2, we make it easier for readers to find the relevant papers and access them. 

Comment 3: Wetting phenomena on structured surface is essential. More detail description will help the readers' understanding. The following article may be suitable as a reference for physical understanding of wettability. 

Hiroyuki Mayama, Wetting Phenomena on Structured Surfaces: Contact Angle, Pinning, Rolling and Bouncing, Biomimetics, Jenny Stanford Publishing https://doi.org/10.1201/9781003277170

Response to Comment 3: We wanted to keep our explanation of wettability as simple as possible to remain accessible to readers from different backgrounds. We did make edits to the introduction in response to some of Reviewer 3’s comments (1 and 2) (lines 47-51, 71-73, 93-101) which provide more detail and clarification. 

We were also unable to access the text suggested (not yet purchased by our library system). If the current revisions are insufficient, we would greatly appreciate further guidance. 

Sincerely,

Marianne Alleyne, Elizabeth Bello, and Yutao Chen

Reviewer 2 Report

This is a nice and interesting review article that deals with hydrophobicity in insects. This theme is wide and deep and so is the manuscript. To make it better understandable for the wide potential readership, the reviewer recommends the following major and minor changes, which need to be implemented to have the manuscript accepted for publication:

1. Add a figure with a sketch of an insect of your choice, and zoom in through the length scales at an area of your interest (e.g., head or leg), pointing out interesting functionalities at each level, and hierarchical dependencies, correlations, going from cm to nm. In this way the complexity of the subject is transferred to the reader in a placative way.

2. Given the wide potential readership across areas and levels of expertise from student to established researcher, from biologist to materials engineer, biomimetics specialist, philosopher of science, chemists, physicists, etc. it is necessary to add a glossary explaining terms that are used in the respective technical wordings of the fields, as well as words that might be used differently in different fields (I as a physicist here think about "quantum leap" which means two different things in physics and in everyday language).

3. there are some nice (but now old) books around that structure similar areas in a nice way (Scherge and Gorb, Biological Micro- and Nanotribology & Shuichi Kinoshita, Structural Colors in the Realm of Nature) - the reviewer recommends to look up these or similar books, refer to them in the references and try to structure the basics of the manuscript in a similar way. What would e.g. be interesting is a table with chemicals used by insects to achieve hydrophobicity, in correlation with the size of the structures on the insects (if any), and also a table which structure types appear on which length scale leading to hydrophobicity.

4. Given the wide and deep focus of this work, it might also be worth discussing with the publisher to add a list of contents at the beginning of the manuscript, with clickable headings, so that firstly, the structure is clear, and secondly, the manuscript becomes more readable.

Author Response

Dear Reviewer 2

We sincerely thank you for providing insightful and constructive comments on our work. We believe that your suggestions have improved the quality and clarity of the manuscript. Based on all 3 reviewers' input we made a few changes to the review paper, including re-naming of headings to help with the organization of the manuscript, further explanation of wettability in the introduction, discussion of droplet jumping, etc.

This is a nice and interesting review article that deals with hydrophobicity in insects. This theme is wide and deep and so is the manuscript. To make it better understandable for the wide potential readership, the reviewer recommends the following major and minor changes, which need to be implemented to have the manuscript accepted for publication:

Comment 1: Add a figure with a sketch of an insect of your choice, and zoom in through the length scales at an area of your interest (e.g., head or leg), pointing out interesting functionalities at each level, and hierarchical dependencies, correlations, going from cm to nm. In this way, the complexity of the subject is transferred to the reader in a placative way.

Response to Comment 1: We hoped to convey in this review the amazing diversity of structures on insect cuticles that result in different functionalities related to wettability. Our Figure 3 attempts to show this diversity in graphical form at different length scales and the description in lines 391-392 provides a narrative on this topic. 

Focusing on just one insect and one body part would not add substantially to the manuscript in our opinion. For example, we have such a figure which we created in the past for presentation purposes. The graphic (attached separately) describes the functionality of nanopillars on cicada wings - the nanopillars result in superhydrophobicity and antimicrobial properties. In the case of nanopillars on cicada wings, hierarchical dependencies have not been reported. If the reviewer thinks that this type of graphic would add to the review then we can hire a graphic designer to create similar graphics. This would delay the publication of the manuscript. 

Comment 2. Given the wide potential readership across areas and levels of expertise from student to established researcher, from biologist to materials engineer, biomimetics specialist, philosopher of science, chemists, physicists, etc. it is necessary to add a glossary explaining terms that are used in the respective technical wordings of the fields, as well as words that might be used differently in different fields (I as a physicist here think about "quantum leap" which means two different things in physics and in everyday language). 

Response to Comment 2: We thank the Reviewer for recognizing the broad appeal of this type of work. However, the journal “Insects” is primarily geared toward entomologists (https://www.mdpi.com/journal/insects/about). We, therefore, do not provide definitions of common entomological terms such as cuticle. We are aware that some of the terminology used in this paper might not be commonly used in other entomological journal articles. To make the paper accessible to entomologists not commonly working in the area of material science we provide detailed descriptions of certain terms in the text, e.g. hydrophobicity, Wenzel and Cassie-Baxter states, nanostructures, microstructures, etc. We also tried to avoid jargon language as much as possible.  

We included Section 2.1 to make sure the reader understood the different terms before moving on to the remainder of the review article. We also did include Table 1 listing synonymous words for certain structures, we felt this was important since no common ontology for insect cuticular structures has been established.

If a glossary is still deemed necessary then we would appreciate a bit more guidance from the Reviewer on what terms should be included, i.e. 1. Common entomological terms (which might not be necessary for an entomology journal), 2. Common material science/engineering/physics terms (which we define in the text), 3. Cuticular structures (for which there currently is no common ontology, an issue that is addressed in the text and Table 1). 

Comment 3. there are some nice (but now old) books around that structure similar areas in a nice way (Scherge and Gorb, Biological Micro- and Nanotribology & Shuichi Kinoshita, Structural Colors in the Realm of Nature) - the reviewer recommends looking up these or similar books, refer to them in the references and try to structure the basics of the manuscript in a similar way. What would e.g. be interesting is a table with chemicals used by insects to achieve hydrophobicity, in correlation with the size of the structures on the insects (if any), and also a table which structure types appear on which length scale leading to hydrophobicity.

Response to Comment 3:

To help with the organization of the article we changed the following headings: 

  1. Introduction → 1. Fundamentals of Wettability in Nature (similar to the Kinoshita-organization)
  2. Materials and Methods → 2. Terminology and Review Methodology (We also moved section 1.4 Overview of Length Scale and Cuticular Terminology into this section so it appears as section 2.1 Overview of Length Scale and Cuticular Terminology and the review methodology is now Section 2.2 Literature Review Methodology (see below)
  3. Terminology and Methodology

2.1. Overview of Length Scale and Cuticular Terminology 

2.2. Literature Review Methodology. 

  1. Results → 3. Hydrophobic Cuticular Structures Found In Insects (Note: We also moved the figures and tables out of this section to where they best fit within the review. Since Table 2 is so large, we moved it to the end of the document). 
  2. Discussion → 4. Discussion and Bioinspired Design Implications
  3. Conclusion → 5. Outlook (similar to the Scherge and Gorb organization)

Regarding the organization of the review article. We outlined many different possibilities for how to outline the review and settled on: an introduction (wettability, hydrophobicity in nature - specifically insects), methodology (brief overview of length scale and cuticle terminology, methods for the literature search), discussion of previous wettability-related results organized by the scale of the structure (=case studies), discussion, and outlook. We decided to organize the results of our literature search by insect taxa within each length-scale section because for material designers who use a bioinspired design framework the taxon is of less importance than length scale and functionality.

We renamed some of the sections to clarify the organization. Some of the heading names were inspired by the Scherge & Gorb and Kinoshita books. Both books are wonderful reviews of important physics topics (tribology, photonics) in nature. However, the organization of a review chapter is a bit more limiting than the organization of a book. 

Organizing the review similar to the Scherge and Gorb book (introduction - review of the cuticular structures found in nature - review of the methodology used to study wettability/hydrophobicity - case studies - conclusions) is not that different from the organization that we proposed, and all the topics are included in our text. Considering that our manuscript is a single review, and not a collection of papers in a book, we think that our organization makes the content accessible. If the reader wants more details they can use the citations within the text, figures, and tables to get to the details. We did not cite this book because most chapters within the book focus on tribological phenomena resulting from cuticular structures, not wettability phenomena. We did, however, cite at least 12 articles on wettability published by the Gorb-group. 

Organizing the review following the wonderful book on structural coloration by Kinoshita would also not be that different from the organization we employed: introduction, fundamental principles of wettability, case studies, and mathematical background. The theory of wettability is not extensively discussed in our paper, however, but we provide the reader with citations to the most pertinent literature on the topic. We did not cite this book specifically since it does not discuss wettability and hydrophobicity, but we do agree that it is an important book on how structure can create multi-functionality. 

Not often is surface chemistry mentioned in relation to hydrophobicity measurements - when it was we included that information in Table 2. Even fewer papers discuss the interplay between physical structure and chemistry, but we do discuss those papers in lines 344-354, 833-836, 862-874.

Rather than creating a table with structure types, length scale, and hydrophobicity values we decided to create a graphic (Figure 4) that compares different structure types at different length scales. The hydrophobicity values are too variable depending on experimental conditions (experimental method, condensation vs evaporative conditions, etc.) to be able to make comparisons between different structures. In addition, many articles on wettability in nature do not actually provide all parameter measurements such as droplet application velocity and droplet size (width and volume), etc.

Comment 4: Given the wide and deep focus of this work, it might also be worth discussing with the publisher to add a list of contents at the beginning of the manuscript, with clickable headings, so that firstly, the structure is clear, and secondly, the manuscript becomes more readable.

Response to Comment 4: Indeed, clickable links would be useful, and probably doable through the “Table of Contents” links in the panel to the left of every “Insects” article. We will be sure to discuss this with the publisher when we get closer to the final publication. Thank you for the suggestion.  

Sincerely,

Marianne Alleyne, Elizabeth Bello, and Yutao Chen

Reviewer 3 Report

This manuscript describes, comprehensively, the various type of hydrophobic, water-repellent natural surfaces. The review is well-written, and the contents were well organized. This review paper can serve as a good guide for researchers to get a comprehensive understanding of the hydrophobic characteristics of insects. However, some minor issues are noticed. I recommend publish this manuscript after these issues being addressed. I have listed them as follow:

1.     Page 2, line 46, the authors states that “Smooth surfaces generally fail to repel water even with intrinsic hydrophobic surface chemistry”. This statement is not accurate. In recent studies, people have developed smooth surface that could repel water or even oil by coating a smooth surface with liquid like polymer brush (Wang, Liming, and Thomas J. McCarthy. "Covalently attached liquids: instant omniphobic surfaces with unprecedented repellency." Angewandte Chemie International Edition 55.1 (2016): 244-248.). I would suggest the authors to rephrase this sentence.

2.     Page 2, line 64, Wenzel state cannot result in superhydrophobicity even though the contact angle could be high. In the field of liquid-repellent surfaces, superhydrophobic surfaces are typically defined as an apparent contact angle of greater than 150 degree AND with contact angle hysteresis of less than 10 degree. There is no discussion on the contact angle hysteresis in the manuscript which is an important parameter that dictates liquid repellency. The authors should refer to the definition of superhydrophobic surfaces and revise the manuscript accordingly (i.e., see for example, D. Quéré, Annu. Rev. Mater. Res. 38: 71-99, 2008).)

3.     Page 2, line 77. Please double check if shark skin is superhydrophobic based on the cited literature.

4.     Figure 2 is not drawn properly for panel c. Note that the maximum contact angle on a flat surface is ~120 degree (see T. Nishino, et al., Langmuir 15: 4321 – 4323, 1999). Any contact angles beyond 120 degree will be attributed to the presence of surface textures related to the Wenzel and Cassie-Baxter discussion.

5.     For Figure 3, adding a needle attaching the droplet will be more realistic.

6.     A few important papers are missing. Please also consider citing the following papers:

a.     Hu, D., Chan, B. & Bush, J. The hydrodynamics of water strider locomotion. Nature 424, 663–666 (2003).

b.     Wang, L., Wang, R., Wang, J., & Wong, T. S. (2020). Compact nanoscale textures reduce contact time of bouncing droplets. Science Advances, 6(29), eabb2307.

Author Response

Dear Reviewer 3 

We sincerely thank you for providing insightful and constructive comments on our work. We believe that your suggestions have improved the quality and clarity of the manuscript. Based on all 3 reviewers' input we made changes to the review paper, including re-naming of headings, further explanation of wettability, discussion of droplet jumping, etc.

Comment 1: Page 2, line 46, the authors state that “Smooth surfaces generally fail to repel water even with intrinsic hydrophobic surface chemistry”. This statement is not accurate. In recent studies, people have developed smooth surfaces that could repel water or even oil by coating a smooth surface with liquid, like polymer brush (Wang, Liming, and Thomas J. McCarthy. "Covalently attached liquids: instant omniphobic surfaces with unprecedented repellency." Angewandte Chemie International Edition 55.1 (2016): 244-248.). I would suggest the authors to rephrase this sentence.

Response to Comment 1: We agree with the reviewer and appreciate the suggestion for how to clarify this issue. We changed the text by adding the following sentence: “Smooth surfaces generally fail to repel water unless accompanied by hydrophobic surface chemistry. The maximum contact angle of water on a chemically treated smooth surface is roughly 120° [2]. In contrast, rough surfaces can exhibit excellent de-wetting properties and enhance hydrophobicity well beyond a 120° contact angle [3–6].” in lines 47-51

Comment 2:    Page 2, line 64, Wenzel state cannot result in superhydrophobicity even though the contact angle could be high. In the field of liquid-repellent surfaces, superhydrophobic surfaces are typically defined as an apparent contact angle of greater than 150 degree AND with contact angle hysteresis of less than 10 degree. There is no discussion on the contact angle hysteresis in the manuscript which is an important parameter that dictates liquid repellency. The authors should refer to the definition of superhydrophobic surfaces and revise the manuscript accordingly (i.e., see for example, D. Quéré, Annu. Rev. Mater. Res. 38: 71-99, 2008).)

Response to Comment 2: We agree with the reviewer and deleted the following sentences: “Both Cassie-Baxter and Wenzel states can result in superhydrophobicity, albeit with differences.​​ Droplets are more likely to exhibit Cassie-Baxter states on rough surfaces, given that the liquid-air interface requires less surface energy [10].” 

We added in the following sentence: ​​”A surface that demonstrates droplets in the Wenzel state may be characterized as hydrophobic, but superhydrophobic characterization only occurs on surfaces with droplets in the Cassie-Baxter state [11].” Line 71-73

We also added the following paragraph in lines 93-101: “Another important parameter is contact angle hysteresis (CAH) which is the difference between the advancing and receding contact angles. Droplets in the Cassie-Baxter state tend to have large advancing and receding contact angles and a low contact angle hysteresis. Droplets in the Wenzel state tend to have a large advancing contact angle, but a low receding contact angle, and thus, a high contact angle hysteresis [11]. It is important to take hysteresis into consideration because two materials may have similar advancing contact angle measurements but different receding contact angles and contact angle hysteresis which drastically affect the adhesion and self-cleaning properties that are usually associated with superhydrophobic materials [5].”

Comment 3: Page 2, line 77. Please double check if shark skin is superhydrophobic based on the cited literature.

Response to Comment 3: We appreciate the reviewer’s comment. We did mistakenly identify shark skin as superhydrophobic. We removed our discussion of shark skin and elaborated on bird feather hydrophobicity (Lines 110-117). 

Comment 4:    Figure 2 is not drawn properly for panel c. Note that the maximum contact angle on a flat surface is ~120 degree (see T. Nishino, et al., Langmuir 15: 4321 – 4323, 1999). Any contact angles beyond 120 degree will be attributed to the presence of surface textures related to the Wenzel and Cassie-Baxter discussion.

Response to Comment 4: Figure 2 is not drawn to demonstrate the impact of smooth and rough surfaces on the maximum contact angle. It serves as a graphical representation for the readers to visualize how apparent contact angle can be used to characterize hydrophilic, hydrophobic, and superhydrophobic surfaces. Many figures in other papers also include a schematic of wettability with a solid surface line that appears to be smooth and flat (see Figure 1 in Evans et al., 2019 PNAS https://doi.org/10.1073/pnas.1913587116, Figure 2 in Zhu et al., 2020 View https://doi.org/10.1002/VIW.20200053, Figure 2 in Samata et al., 2020 Materials and Design, doi:10.1016/j.matdes.2020.108744,  Figure 1 in Cirisano et al., 2021 Coatings https://doi.org/10.3390/coatings11121508). 

We slightly altered the image to emphasize the contact angle and altered the caption to read “Figure 2. Schematic representation of wettability characterization (a) hydrophilic contact angle (CA) less than 90° (b) hydrophobic CA greater than 90° (c) superhydrophobic CA greater than 150°. Note: This diagram is not designed to show surface roughness beneath the droplets.” (Lines 74-75)

Comment 5: For Figure 3, adding a needle attaching the droplet will be more realistic.

Response to Comment 5: After some discussion, we decided to remove Figure 3 since the needle-and-drop method is only one way to measure contact angle hysteresis. In our lab, we actually use a microgoniometer (Kyowa MCA-3) equipped with an inkjet head that uses piezoelectric technology to dispense droplets (without the use of a needle). 

Comment 6: A few important papers are missing. Please also consider citing the following papers:

  1. Hu, D., Chan, B. & Bush, J. The hydrodynamics of water strider locomotion. Nature 424, 663–666 (2003).

Response to Comment 6a: We did not include the Hu et al. paper because that paper does not discuss or show images of the cuticular structures or contact angle measurements. The paper mentions that the leg hairs are super-hydrophobic and enable the hydrodynamic phenomena that prevent water striders from breaking the surface tension while also moving across the surface. The Hu manuscript refers to an Andersen article (1976,) and a de Gennes et al. book (2002, in French) as the sources for measurements that show that the hairs are hydrophobic. The Andersen article mainly discusses waterstrider locomotion, there is some mention of hair structures that exist on the legs, but no contact angle measurements are provided. Since neither the Hu et al. paper (2003) nor the Andersen paper (1976) quantifies hydrophobicity we feel that they should not be included in this review.  We have requested the de Gennes book, and translation services, from InterLibrary Loans and will add these citations to later versions of our review after we are able to determine that these sources provide the data that supports that water strider legs are hydrophobic. We are not sure if we will be able to do that before publication. However, we feel this is not a major issue since the cited Gao and Jiang (2004) paper provides us with information on hairs (SEM micrographs) and contact angle measurements that fit the scope of this review.  

  1. Wang, L., Wang, R., Wang, J., & Wong, T. S. (2020). Compact nanoscale textures reduce contact time of bouncing droplets. Science Advances, 6(29), eabb2307.

Response to Comment 6b: We thank the reviewer for this suggestion. After reading the article we have incorporated it into the concluding section of our paper in lines 845-859: “Historically, based on the Cassie-Baxter theory, low solid fraction textures were deemed essential in the creation of water repellent materials. However, a recent study found that certain insects with high solid fraction textures are still able to achieve water repellency by reducing the texture size below 300 nm [108]. Additionally, this study postulated that the compact and nanoscale features of insect surfaces work favorably to rapidly shed high-impact water droplets like rain [108], thus maybe even carrying away contaminating particles from a surface [109]. Passive self-cleaning has been shown to occur in nature through coalescence-induced jumping of microscale condensate droplets during fogging, dewing, or condensation, including in insects [110], even when just one droplet coalesces with a particulate of a certain size under those conditions [111]. Particle transport physics was shown to depend on the wettability characteristics of surfaces, including natural surfaces, e.g. butterfly wings, cicada wings, and clover leaves  [111]. Studies like these on rapid droplet shedding and droplet jumping, including particle-droplet jumping, can provide guidelines for the design of artificial water-repellent and passive self-cleaning surfaces. 

Sincerely,

Marianne Alleyne, Elizabeth Bello, and Yutao Chen

Round 2

Reviewer 2 Report

The authors implemented my recommendations or gave satisfactory explanation why not in the cases where they did not, so I recommend publication.